# High Diet Quality Is Linked to Low Risk of Abdominal Obesity among the Elderly Women in China

**DOI:** 10.3390/nu14132623

**Published:** 2022-06-24

**Authors:** Lixin Hao, Hongru Jiang, Bing Zhang, Huijun Wang, Jiguo Zhang, Wenwen Du, Chunlei Guo, Zhihong Wang, Liusen Wang

**Affiliations:** National Institute for Nutrition and Health, Chinese Center for Disease Control and Prevention, 29 Nanwei Road, Beijing 100050, China; haolx@ninh.chinacdc.cn (L.H.); jianghr@ninh.chinacdc.cn (H.J.); zzhangb327@aliyun.com (B.Z.); wanghj@ninh.chinacdc.cn (H.W.); zhangjg@ninh.chinacdc.cn (J.Z.); duww@ninh.chinacdc.cn (W.D.); gclwyf@163.com (C.G.); wangzh@ninh.chinacdc.cn (Z.W.)

**Keywords:** dietary guideline index, diet quality, elderly, obesity, China

## Abstract

How diet as a whole impacts the risk of general overweight and abdominal obesity among the Chinese elderly is unclear. The present study aimed to examine the association of overall diet quality with general overweight and abdominal obesity in the Chinese elderly. Using data from the China Health and Nutrition Survey (CHNS) from 1993 to 2015, an ongoing cohort study, we selected participants aged 60 and older who were not generally overweight, but who had abdominal obesity at baseline and who had participated in at least two waves of the survey as subjects. The China Elderly Dietary Guidelines Index (CDGI-E) was used, based on the critical diet-related recommendations of the 2016 Chinese Dietary Guideline (CDG-2016), to assess overall diet quality. Consecutive 3 d, 24 h recalls and household weighing for seasonings and edible oils were used to collect dietary data and calculate the CDGI-E scores. Three-level (community-individual-wave) random intercept logistic regression models were used to analyze the impact of diet quality on the risk of general overweight and abdominal obesity in the elderly. The results showed that the older women in the top sixtiles of the CDGI-E scores had a 38% reduction —0.62, 95% CI (0.41, 0.92)—in the risk of abdominal obesity, as compared to those in the bottom sixtiles after adjusting for all potential confounders, while the null association was observed in the older men. The relationship between CDGI-E score and the risk of overweight/general obesity in the Chinese elderly has not been found. It was concluded that a high diet quality was associated with a reduced risk of abdominal obesity among elderly women in China. Our findings will help to improve the understanding of the relationship between the overall effect of diet and health. It may provide a new avenue for obesity intervention policy formulation from the aspect of improving overall dietary quality.

## 1. Introduction

Cardiovascular disease (CVD) is the leading cause of death for people over 50 years old [1]. With the development of the social economy, the incidence of cardiovascular disease among Chinese residents is also increasing, and it takes first place among the total causes of death in residents [2]. Several epidemiologic studies had proved that general overweight and abdominal obesity, as cardiometabolic risk factors, can increase the risk of diabetes, hypertension, and cardiovascular diseases [1,3]. Over the past two decades, the prevalence of general overweight and abdominal obesity has risen dramatically worldwide, as seen in China [4,5,6,7]. A healthy diet is closely related to the physical health of humans; it also proves the association between diet and health outcomes [7,8].

At present, the food environment is changing rapidly. People eat out and eat processed food more frequently. Malnutrition, such as excessive intake of macronutrients, lack of micronutrients, and food insecurity, persist [9]. There are some dietary problems that are frequently observed in the elderly, such as the excessive intake of energy, saturated fat, and meat, and the insufficient intake of protein and micronutrients [10]. An American study shows that from 2001 to 2018, the dietary quality of the elderly gradually decreased [11]. The nutritional imbalance of the elderly in China is also prominent, which is mainly manifested by the excessive intake of meat and edible oil, but the insufficient intake of vegetables, fruits, soybeans, and dairy products [12]. Some studies show that dietary energy density, the consumption of processed food, red meat intake, and other dietary factors were associated with the prevalence of obesity, cardiovascular diseases, type 2 diabetes, etc. [13,14,15].

It is known that dietary patterns can preferably evaluate the relationship between diet and obesity in more than a single type of food [16]. A better understanding of the link between diet quality and health outcomes is of such importance that it motivates policymakers make informed decisions to curb the problem of diet-induced health-related issues. Some studies suggested that satisfactory Healthy Eating Index, Alternate Healthy Eating Index, Mediterranean Diet, or Dietary Approaches to Stop Hypertension (DASH) diet scores were associated with a lower risk of general overweight and abdominal obesity in adults and children in the US, Northern Europe, Iran, and China [17,18,19,20,21]. Moreover, one recent study showed that Chinese diet quality accessed by CDGI-2007 was negatively related to the risk of high LDL-C and insulin resistance [22,23]. However, few studies have examined the association between diet quality and obesity in the Chinese population.

Therefore, for this study, we used the China Elderly Dietary Guidelines Index, referring to the dietary guidelines for the elderly in the DGC-2016 and CDGI-2007. Diet quality was calculated using dietary data collected from the China Health and Nutrition Survey (CHNS). This study was designed to examine the relationship between diet quality assessed by the CDGI-E and the risk of general overweight and abdominal obesity among Chinese residents, aged above 60 years old, from 1993 to 2015.

## 2. Materials and Methods

### 2.1. Study Population and Design

The CHNS, an ongoing series of longitudinal studies initiated in 1989, established by the National Institute for Nutrition and Health, the Chinese Center for Disease Control and Prevention, and the University of North Carolina, focused on the impact of the social and economic transformation of Chinese society on the health and nutritional status of its population. Initially, the CHNS involved 8 provinces (regions). By 2015, the development of the research included 15 provinces (regions/municipalities). A multi-stage cluster random sampling method was used to select communities, randomly selecting 20 households in each community and investigating all family members. The details of the sampling process can be found elsewhere [24,25]. The CHNS has completed 10 rounds (1989, 1991, 1993, 1997, 2000, 2004, 2006, 2009, 2011, and 2015) of the study. The survey includes household (individual activities, lifestyle, health status, marriage and birth history, body shape, mass media exposure, etc.), physical measurement (waist circumference, hip circumference, triceps skin fold, etc.), diet (food and condiment), and community (community infrastructure, commodity price, health services, etc.). It collected waist circumference (WC) for the first time in 1993.

For the present study, we used data from eight waves of the survey, between 1993 and 2015, due to the waist circumference being measured in 1993. We selected the subjects who were 60 years old and above and who had participated in at least two waves of the survey, with complete diet, demographic, socioeconomic, and anthropometric data. Through data cleaning, the subjects who had very low or very high energy intakes (<800 or >6000 kcal per day in men and <600 or >4000 kcal per day in women), extreme WC (<50 or >130 cm), or extreme BMI (<13.9 or >61.5 kg/cm^2^) were deleted. Then, we excluded participants with a baseline history of hypertension, diabetes, stroke, or myocardial infarction, as well as those who had only participated in the survey once. The final analysis included 1965 observations for the study of abdominal obesity (removed baseline abdominal obesity) and 1710 observations for the study of overweight/general obesity (removed baseline overweight/general obesity).

All procedures involving human subjects of the CHNS were approved by the Ethics Committee of the National Institute for Nutrition and Health, and the Chinese Center for Disease Control. All subjects provided written informed consent for their participation in the protocols (2015017).

### 2.2. Construction of the CDGI-E

The CDGI-2007 comprised 6 adequacy components and 4 moderating components. All components were scored at successive points from 0 to 10. The total points ranged from 0 to 100. The Chinese residents’ dietary reference intake was used as a standard for dietary assessment. The scoring system has been elaborated elsewhere [23].

The DGC-2016 is established based on the scientific principles of nutrition and the health needs of residents. This guideline includes six qualitative core items involving diet, physical activity (PA), healthy weight, alcohol intake, and eating habits. Meanwhile, the revised Chinese Food Guide Pagoda (CFGP) lists 12 kinds of foods, with quantitative recommendations [19]. Energy requirements for the elderly of different ages and genders are identified according to the China Nutrition Society dietary intake recommendations [26] (Table 1).

For exploring the independent relationship between diet quality and obesity, we used the CDGI-E based on the dietary recommendations. Based on the CDGI-2007, we replaced the coarse grains with other grains and dry beans and added the percentage of energy from carbohydrates and eggs. Meanwhile, the recommendation of corresponding energy levels for the elderly (2000−2200 kcal for males and 1600−1800 kcal for females) were selected based on the new edition of the DGC-2016 and the Chinese residents’ dietary reference intake. The components, the criteria for the highest and lowest scores, and the computing methods for the middle scores are shown in Table 2. The CDGI-E, which ranges from 0 to 110 points, consists of three major categories, including adequate components (cereals and tubers, including the percentage of energy from carbohydrates and other grains and dry beans; fruit; vegetables, including vegetable intake and the quantity of dark-colored vegetables; soybeans, and nuts; and dairy products), moderate components (red meat, poultry, eggs, and seafood), and limited components (edible oil, salt, and alcohol). The higher the score, the better the DGC-2016 compliance.

Use the following formula to compute the middle score of each component: For adequate components, component score = highest score/{[(R_max_ − R_min_)/2] + R_min_} × X. However, the scoring rule for the percentage energy from carbohydrates is an exception, it follows the scoring rule for moderate components. When the proportion is lower than 50%, the component score = highest score/R_min_ × X; when the proportion is higher than 65%, the component score = highest score/R_max_ × (X − R_min_). For limited components, when the consumption is lower than the limit, the component score = highest score − highest score/R_max_ × (X − R_max_).

R_max_ is the maximum recommendation for the corresponding component, R_min_ is the minimum recommendation, and X is the actual personal intake of each component.

### 2.3. Dietary Quality Measurement

Consecutive 3 d (two weekdays and one weekend day), 24 h recalls were used to collect information on all foods and beverages, consumed at home and away from home, for each subject by trained interviewers. The 3 d household weighing method was used to collect the consumption of condiments and edible oils, which were divided among each family member according to their dietary energy intakes. Edible oil includes vegetable oil and animal fat, which are used in cooking. Further description of the collection method was recorded elsewhere [25,27]. We then used the average intake of the 3 days for each individual. The food intake data were linked to the China Food Composition Table (FCT) to calculate the intake of energy and nutrients and the percentage of energy from carbohydrates [28]. The percentage of energy from carbohydrates and foods/food groups was used to calculate the CDGI-E in the analyses. The total score of the CDGI-E equals the sum of each component score, with a higher score indicating higher diet quality. We divided the participants into sixtiles by wave and gender, giving the statistically significant interaction between gender and the CDGI(2018)-E on the risk of general overweight and abdominal obesity.

### 2.4. Assessment of Obesity

Abdominal obesity: At each visit, the individual WC was measured as midway between the lowest rib and the iliac crest by trained health workers using a SECA tape measure. Abdominal obesity is defined as a WC of ≥90 cm for men and ≥80 cm for women, according to the International Diabetes Federation (IDF) definitions for the Asia population [29].

Overweight/general obesity: Height was measures with a reel-type height gauge (SECA206, Saikang, Hamburg, Germany), accurate to 0.1 cm. Body weight was measured using a fat measuring instrument (TANITABC601, Berida, Tokyo, Japan), accurate to 0.1 kg. Measurements were taken twice in a row, and the average value was obtained. BMI was calculated as body weight (kg)/square height (m^2^). Overweight/general obesity was defined as BMI ≥ 25 kg/m^2^, by the Asia-Pacific criteria of the WHO guidelines [30].

### 2.5. Assessment of Relevant Covariates

Detailed information on the participants’ sociodemographic and lifestyle factors was collected by trained investigators at each wave, using standard questionnaires. the questionnaires included the participant’s age, education level, physical activity, smoking status, annual household income, and disease history (hypertension, diabetes, stroke, and myocardial infarction).

Education level was categorized as less than primary school, primary school, middle school, and advanced. The provinces were grouped into central, east, and west by geographical regions. Physical activity (PA)–metabolic equivalent of task (MET)–was estimated based on the standard questionnaires (including occupational, household chores, leisure time, and transportation activities), and total MET-hours per week were calculated, accounting for both the average intensity of each activity and the time spent in each activity, and then were categorized into light, moderate, and heavy [31]. Annual household income was grouped into three parts by rank. The community urbanicity index is a complex index to measure the degree of urbanization. It was established by 12 components, including population density, economic activity, markets, infrastructure, social services, and so on, at the community level [32]. The community urbanicity index was categorized into trisection. Smoking status was divided into ever a smoker and nonsmoker. The analysis also considers baseline WC as a potential confounder.

### 2.6. Statistical Analysis

The baseline characteristics of the subjects were presented as the median for continuous and non-normally distributed variables and the proportion for categorical variables across sixtiles of the CDGI-E scores. Because the dietary quality of most people in the study was low, to better compare the effects of different dietary qualities on obesity, we divided the participants into sixtiles, instead of the traditional quartiles, according to CDGI-E score and gender. We performed the Kruskal–Wallis tests for the CDGI-E scores, age, energy intake, BMI (body mass index), and WC; chi-square tests were performed for PA levels, educational levels, geographic regions, urbanicity index, household income, and smoking status [33].

The association between obesity and CDGI-E score was assessed using a three-level random intercept logistic regression model, including general overweight and abdominal obesity. Firstly, a null model, comprising waves (level 1), individuals (level 2), and communities (level 3), was created, with none of the predictor variables. Then, we added the CDGI-E score and adjusted for age, educational level, geographic region, urbanicity index, household income, and energy intake in Model 1. PA and smoking status were further adjusted in Model 2. Model 3 introduced the baseline WC. The median values were assigned to sixtiles of the CDGI-E score and used to fit the model as a continuous term for testing linear trends. All statistical analyses were conducted using SAS version 9.4 (SAS Institute Inc., Cary, NC, USA) and Stata version 15.0 (Stata Corp., College Station, TX, USA).

## 3. Results

### 3.1. Characteristics of Participants

A total of 1965 elderly (men 60.6%, women 39.4%), with completed data, were included at the baseline (including overweight/general obesity, excluding abdominal obesity). The average age of the subjects was 64.88 ± 5.42 years old, ranging from 60.00 to 93.83 years old. At the baseline, the subjects’ BMI was 21.07 ± 2.62 kg/cm^2^ and waist circumference was 75.25 ± 7.04 cm. The elderly having higher CDGI-E scores tended to be those with a higher household income, higher education level, and residence in a more highly urbanized community. For men, higher scores were positively related to a lower proportion of smoking. The women with higher scores tended to live in the east region (Table 3). At the endpoint, the abdominal obesity rate was 30.68%, and the overweight/general obesity rate was 13.03% (Appendix A).

### 3.2. Association between Overall Diet Quality and Overweight/General Obesity

Table 4 shows the three-level mixed-effects logistic regression results of the association between the CDGI-E score and the risk of overweight/general obesity in the Chinese elderly. The relationship between CDGI-E score and the risk of overweight/general obesity in the Chinese elderly has not been found. No significant linear trends were observed.

### 3.3. Association between Overall Diet Quality and Abdominal Obesity

Table 5 shows the three-level mixed-effects logistic regression results of the association between the CDGI-E score and the risk of abdominal obesity in the Chinese elderly. Overall, the CDGI-E score is inversely associated with the risk of abdominal obesity only in women; that is, the odds ratio (OR) was 0.61 (95% CI, 0.41 to 0.91) after adjusting for demographic and economic confounders; the OR was 0.60 (95% CI, 0.40 to 0.89) after additionally adjusting for individual healthy behavior (smoking and physical activity); and the OR was 0.62 (95% CI, 0.41 to 0.92) after additionally introducing the baseline WC. No significant linear trends were observed.

## 4. Discussion

In the present study, we used the established CDGI-E, based on the diet-related recommendations of DGC-2016, to evaluate the diet quality of the Chinese elderly. In addition, we examined the association with overweight/general and abdominal obesity in elderly people above 60 years old in China. We found that the overall diet quality of the Chinese elderly was unsatisfactory. There were statistically significant differences in the CDGI-E at different levels of PA, education, household income, urbanicity index, and smoking status. The top sixtiles in CDGI-E score were associated with an 8 to 59% reduction in the risk of abdominal obesity in women, compared with the bottom sixtiles.

There are two main approaches to the study of dietary patterns: the data-driven dietary pattern, based on factor or cluster analysis, and the index-based dietary pattern. The latter is used to establish a simple, practical, and comprehensive index based on the recommendations of food and/or nutrient intake from the dietary guidelines or other evidence-based scientific guidance, and the index-based diet quality measurement given as a summary score is widely used in various studies. The composition of the three-diet index mentioned above was based on the corresponding dietary guideline, respectively. In China, the China Dietary Guideline Index (CDGI-2007) was used, based on DGC-2007. It consists of 12 components, in correlation with diet quality and cardiometabolic risk factors. The revised DGC was published in 2016 [34]. Subsequently, HE Yu-na et al. designed the China Healthy Diet Index (CHDI) for evaluating the diet quality of Chinese adults [35]. Compared with CDGI-2007, it added food variety, refined grains, calories from saturated fatty acid, and empty calories, and replaced salt with sodium. The CHDI added more nutrient components for greater precision in evaluating diet quality, but the index is relatively complex. Meanwhile, the Chinese Healthy Eating Index (CHEI) was created by Ya-Qun Yuan et al. and was applied to the general population [36]. The CHEI included 17 components, both comprehensive and specific, but it is necessary to narrow the scoring range of some components. The DGC-2016 consisted of evidence-based scientific guidance for the establishment of the CDGI-E and the CDGI-E score aimed to quantify, to some extent, the adherence to diet-related recommendations of the DGC-2016 among the Chinese elderly people. Compared with the existing diet index, the CDGI-E has several advantages. Firstly, no other index has been previously constructed specifically to evaluate the diet quality of the elderly in China, although other countries have developed diet indexes for older people and further applied them in the studies on diet-disease relationships [37,38,39]. Secondly, the CDGI-E has stronger applicability, since Chinese dietary habits are different from those in other countries. Thirdly, the CDGI-E, established as a consecutive scale, can assess diet quality dynamics more effectively and predict disease risk more precisely than the indexes used as discontinuous scores [40].

Whether in developed or developing countries, the diet quality of human beings is increasingly improving [41,42,43]. Our results supported the view that individuals with higher educational levels are more likely to have high-quality diets [44], and the view that non-smokers are likely to follow DGC recommendations [45]. It is slightly unexpected that the elderly with high PA levels had lower diet quality scores than those with light PA levels, as opposed to the results from previous studies among the Chinese adults [22,23]. It may be that the elderly with heavy PA levels are more focused on energy-based food intake and pay less attention to other types of food.

There are few studies on the connection between the dietary quality of the elderly and abdominal obesity. In line with the results from other countries, our study also observed the inverse association of the CDGI-E scores with the risk of abdominal obesity among older women, after adjusting for all potential confounders [16,17,18,19]. Such results reflected the practicability of the new scoring approach in predicting the risk of abdominal obesity. However, we only found a threshold effect of diet quality by comparing the top and bottom sixtiles of the scores, rather than a significant linear trend. It is most likely that the diet quality could reduce the risk of abdominal obesity, to a certain degree. The median of the top sixtiles CDGI-E score is 59.77 points. Our study showed that the diet quality of the Chinese elderly is still relatively low despite some improvement over time. Limited variation of the score and a small proportion of subjects having high diet quality made it impossible to do a more in-depth analysis. At present, most studies believe that dietary index is associated with overweight and obesity [46,47,48], but some studies have not found this association [49]. Our study did not find the impact of dietary quality on overweight/general obesity. This may be due to the insufficient representativeness of the samples and the authenticity of the results, which needs to be further explored by expanding the sample size in the future.

Compared with the research on other age groups, the overweight and obese elderly do not pay enough attention to diet quality and health-related issues, which hinders the formulation of nutrition policies for obesity in the elderly. A systematic review showed that only 1 of 479 articles was applied to the elderly, and very few studies have been followed up for more than ten years [50]. The strengths of our research include the prospective design, large sample sizes of a special population, precise dietary data collection using consecutive 3 d, 24 h recalls, combined with household weighing for cooking oils and condiments, the use of CDGI-E to measure overall diet quality and its association with abdominal obesity, the use of a multilevel mixed-effect model to handle the hierarchal and unbalanced data structure, allowing non-independence of repeated measurement, and adjustment for comprehensive latent covariables and potential heterogeneity. This study was followed up for more than 20 years to reflect the long-term effect of dietary patterns on body weight. The CDGI-E was used, based on the critical diet-related recommendations of the CDG-2016, to access overall diet quality, which is more suitable for Chinese people. To partly control the potential effect of reverse causation on the findings, we removed individuals with diet-related chronic disease at baseline. Previous studies have shown that this effect is more obvious in women, but not in men. Even those with persistent abdominal obesity seem to underestimate the seriousness of the problem, and did not experience more control over their weight [51]. This situation is worrying, but it also proves, to some extent, that the reverse causality has less influence. Moreover, the prospective study has the advantage that it can prove a more time-dependent relationship between the nature status and the occurrence of disease.

Instead of the CDS-2013 established by the Chinese Diabetes Society for judging abdominal obesity (WC of ≥90 cm for men and ≥85 cm for women) [52], we selected the standard of IDF-2005, which may be more sensitive in women (WC of ≥90 cm for men and ≥80 cm for women) [29]; in a future study, we will compare the results using two definitions. In this study, 68.75% of the elderly are under 65 years old, and only 0.94% of the elderly have a BMI ≥ 30.0. Studies have shown that BMI > 25 in the elderly will also increase the risk of type 2 diabetes, cardiovascular diseases, and cancer [10]. To better compare the influence of dietary quality on BMI, we have chosen the criteria of the WHO Guidelines for adults, in which BMI ≥ 25.0 is defined as overweight/general obesity [30].

The study also has certain limitations. Firstly, the change of diet patterns or habits is unavoidable when the participants were diagnosed with an acute or chronic disease, and this might influent the evaluation of diet quality, resulting in the deviation of the study. For reducing bias as much as possible, we excluded the elderly with histories of diseases that would most likely change dietary structure, but since so many individuals living in the community have chronic case histories, this surely limited the study’s representativeness and generalizability. Secondly, due to the index scores being mostly distributed in the range of the lower scores, we divided the participants into sixtiles of practical scores. This might underestimate the association between diet quality and obesity. Thirdly, residue confounding may exist, due to the nature of the observational study, although we have adjusted many related covariables. Moreover, consecutive 3 d, 24 h dietary recalls have limited ability to capture the intake of sporadically consumed foods. Moreover, this type of data cannot reflect seasonal differences in dietary quality. But the average intake for 3 days can provide a relatively precise assessment of usual diet quality, especially using two weekdays and one weekend day, which we selected considering the potential difference in diet between weekdays and weekend days.

## 5. Conclusions

The diet quality of Chinese elderly women, measured by the CDGI-E, was independently associated with the risk of abdominal obesity. Therefore, attention should be paid to the dietary quality of the elderly, especially that of elderly women. It is necessary to strengthen the research on the dietary factors related to overweight and obesity in the elderly, to provide the scientific basis for dietary intervention. It is suggested that the elderly should be fully considered in the formulation of obesity prevention and control policies. Efforts should be made to expand science popularization, education, and nutrition interventions aimed at improving the dietary quality of the elderly. Further studies should be conducted to explore the CDGI-E’s effect on other cardio-metabolic risk factors for verifying its practicability.

## Figures and Tables

**Table 1 nutrients-14-02623-t001:** Balanced dietary patterns and amounts of food at different energy levels (g/d).

Varity of Food	Energy Intake Level (kcal)										
	1000	1200	1400	1600	1800	2000	2200	2400	2600	2800	3000
Grain	85	100	150	200	225	250	275	300	350	375	400
Whole grain and beans	Moderate			50–150							
Tuber	Moderate			50–100					125	125	125
Vegetable	200	250	300	300	400	450	450	500	500	500	600
Dark-colored vegetable	Accounts for half of all vegetables										
Fruit	150	150	150	200	200	300	300	350	350	400	400
Red meat and poultry	15	25	40	40	50	50	75	75	75	100	100
Egg	20	25	25	40	40	50	50	50	50	50	50
Seafood	15	20	40	40	50	50	75	75	75	100	125
Dairy products	500	500	350	300	300	300	300	300	300	300	300
Soybeans	5	15	15	15	15	15	25	25	25	25	25
Nuts		Moderate		10	10	10	10	10	10	10	10
Cooking Oil	15–20	20–25			25	25	25	30	30	30	35
Salt	<2	<3	<4	<6	<6	<6	<6	<6	<6	<6	<6

The energy range of the Chinese Food Guide Pagoda is 1600–2400 kcal. Potatoes are measured according to fresh weight.

**Table 2 nutrients-14-02623-t002:** Components of the CDGI-E and scoring methods, according to the DGC and CFGP.

Qualitative Recommendations of DGC	Quantitative Recommendations of CFGP	Components of CDGI-E	Recommendation for Elderly ^b^	Criteria for Lowest Score (0) ^d^	Criteria for Highest Score ^d^	Highest Score Value
Eat a variety of foods, cereal based.	Grains, beans, and tubers: 250–400 g/d	Percentage of energy from carbohydrates	50−65%	0% or 100%	50−65%	5
	Whole grains and beans: 50–150 g/d	Other grains and beans	50–150 g/d	0 g/d	≥100 g/d	5
	Tubers: 50–100 g/d					
Eat plenty of vegetables, fruits, dairy products, and soybeans.	Vegetables: 300–500 g/d	Vegetables	Male: 450 g/d	0 g/d	Male: ≥450 g/d ^a^	5
			Female: 300–400 g/d	0 g/d	Female: ≥350 g/d ^a^	
		Dark-colored vegetables ^c^	>1/2	0	≥1/2	5
	Fruits: 200–350 g/d	Fruits	Male: 300 g/d	0 g/d	Male: ≥300 g/d ^a^	10
			Female: 200 g/d		Female: ≥200 g/d ^a^	
	Dairy products: 300 g/d	Dairy products	300 g/d	0 g/d	≥300 g/d	10
	Soybeans and nuts: 25–35 g/d	Soybeans and nuts	Male: 25–35 g/d	0 g/d	Male: ≥30 g/d ^a^	10
			Female: 25 g/d	0 g/d	Female: ≥25 g/d ^a^	
Eat a moderate amount of fish, poultry, eggs, and lean meats.	Seafood: 40–75 g/d	Seafood	Male: 50–75 g/d	0 g/d	Male: ≥62.5 g/d ^a^	10
			Female: 40–50 g/d		Female: ≥45 g/d ^a^	
	Red meat and poultry: 40–75 g/d	Red meat and poultry	Male: 50–75 g/d	Male: 0 g/d or ≥125 g/d	Male: 62.5 g/d ^a^	10
			Female: 40–50 g/d	Female: 0 g/d or ≥90 g/d	Female: 45 g/d ^a^	
	Eggs: 40–50 g/d	Eggs	Male: 50 g/d	0 g/d	Male: 50 g/d	10
			Female: 40 g/d	0 g/d	Female: 40 g/d	
Limit salt, cooking oil, added sugar, and alcohol.	Edible oil: 25–30 g/d	Edible oil	25 g/d	50 g/d	25 g/d	10
	Salt: <6 g/d	Salt	<6 g/d	≥12 g/d	<6 g/d	10
		Alcohol	Male: <25 g/d	Male: ≥50 g/d	Male: <25 g/d	10
			Female: <15 g/d	Female: ≥30 g/d	Female: <15 g/d	

Abbreviation: CDGI-E, China Elderly Dietary Guidelines Index; DGC, Dietary Guidelines for Chinese; CFGP, Chinese Food Guide Pagoda. ^a^ Maximum is the median of recommended range. ^b^ Recommendations of components intake in different energy requirements. (see Table 1). ^c^ Dark-colored vegetables are defined as 500 mg carotene/100 g of vegetables. ^d^ For adequate components, score = highest score/{[(R_max_ − R_min_)/2] + R_min_} × X. However, the scoring rule for percentage of energy from carbohydrates is an exception; it follows the scoring rule of the moderate components. When the proportion is lower than 50%, component score = highest score/R_min_ × X; when the proportion is higher than 65%, score = highest score/R_max_ × (X − R_min_). For limited components, when the consumption is lower than the limit, component score = highest score − highest score/R_max_ × (X − R_max_). R_max_ is the maximum recommendation of the corresponding component, R_min_ is the minimum recommendation, and X is actual personal intake of each component.

**Table 3 nutrients-14-02623-t003:** Baseline characteristics of participants by gender in CHNS.

Characteristics	Men						*p* ^2^	Women						*p* ^2^
	S1 (*n* = 224)	S2 (*n* = 194)	S3 (*n* = 188)	S4 (*n* = 202)	S5 (*n* = 188)	S6 (*n* = 195)		S1 (*n* = 119)	S2 (*n* = 130)	S3 (*n* = 133)	S4 (*n* = 132)	S5 (*n* = 119)	S6 (*n* = 141)	
CDGI-E scores ^1^	25.93	32.93	37.57	42.55	47.86	56.44	<0.001	29.33	35.91	41.60	45.98	50.67	59.93	<0.001
Age (y) ^1^	62.40	62.61	62.33	62.68	63.06	63.09	0.040	62.79	62.46	62.93	62.90	62.55	63.99	0.048
PA (%)														
Light	19.20	19.59	15.43	19.31	26.60	30.77	<0.001	9.24	12.31	15.04	16.67	11.76	12.77	0.003
Moderate	24.55	21.13	31.38	29.21	23.40	31.28		31.93	30.77	31.58	28.79	37.82	51.06	
Heavy	56.25	59.28	53.19	51.49	50.00	37.95		58.82	56.92	53.38	54.55	50.42	36.17	
Educational level (%)														
Less than primary school	46.88	48.45	40.96	45.05	42.55	41.54	0.012	77.31	82.31	81.20	81.06	73.11	58.87	<0.001
Completion of primary school	30.36	22.68	29.26	27.23	27.66	18.97		17.65	12.31	8.27	11.36	15.13	23.40	
Middle school or advanced	22.77	28.87	29.79	27.72	29.79	39.49		5.04	5.38	10.53	7.58	11.76	17.73	
Geographic region (%)														
Central	46.43	36.60	43.62	40.59	38.30	33.85	0.623	41.18	42.31	34.59	35.61	18.49	26.24	0.004
East	24.11	26.29	22.87	24.75	30.85	41.54		25.21	18.46	23.31	16.67	29.41	39.01	
West	29.46	37.11	33.51	34.65	30.85	24.62		33.61	39.23	42.11	47.73	52.10	34.75	
Urbanicity index (%)														
Low	44.64	40.72	35.11	31.19	28.19	20.51	<0.001	48.74	40.77	41.35	39.39	26.05	19.86	<0.001
Middle	32.59	33.51	32.98	34.65	35.64	26.67		32.77	30.00	32.33	37.12	40.34	26.24	
High	22.77	25.77	31.91	34.16	36.17	52.82		18.49	29.23	26.32	23.48	33.61	53.90	
Household income (%)														
Low	39.29	38.66	34.04	29.70	30.32	22.56	<0.001	47.06	40.77	39.10	37.12	26.89	29.08	<0.001
Middle	32.59	29.90	32.98	29.70	34.04	28.72		28.57	30.00	31.58	33.33	31.93	25.53	
High	28.13	31.44	32.98	40.59	35.64	48.72		24.37	29.23	29.32	29.55	41.18	45.39	
Ever smokers (%)	66.96	69.59	63.30	62.87	63.30	58.46	0.030	7.56	10.00	9.77	12.12	9.24	14.89	0.096
Energy intake (kcal/day) ^1^	2590.52	2482.68	2294.73	2187.27	2203.64	2099.76	<0.001	2132.65	2149.74	1980.12	1941.63	1920.70	1784.00	<0.001
BMI (kg/cm^2^) ^1^	20.94	20.85	21.36	21.51	21.23	21.64	0.127	20.43	20.47	19.70	20.30	20.50	20.40	0.172
WC (cm) ^1^	77.35	77.00	77.70	78.00	78.00	79.00	0.826	73.00	73.00	73.00	72.00	72.00	73.00	0.572

Abbreviation: CDGI-E, China Elderly Dietary Guidelines Index; S, sixtile; PA, physical activity; BMI, body mass index; WC, waist circumference; CHNS, China Health and Nutrition Survey. ^1^ Median. ^2^ Wilcoxon rank-sum test for continuous and non-normal distributed variables, and chi-square test for categorical covariates.

**Table 4 nutrients-14-02623-t004:** OR (95% CI) of the risk of overweight/general obesity across the CDGI-E scores.

	Model 1 ^2^	Model 2 ^2^	Model 3 ^2^
Fixed effect			
CDGI-E scores ^1^			
S1	1.00 (1.00, 1.00) ^a^	1.00 (1.00, 1.00) ^a^	1.00 (1.00, 1.00) ^a^
S2	1.05 (0.61, 1.79)	1.07 (0.62, 1.83)	1.02 (0.60, 1.74)
S3	1.06 (0.63, 1.78)	1.07 (0.63, 1.80)	1.05 (0.62, 1.75)
S4	1.12 (0.66, 1.92)	1.13 (0.66, 1.93)	1.05 (0.62, 1.79)
S5	1.02 (0.60, 1.73)	1.02 (0.60, 1.74)	0.99 (0.58, 1.69)
S6	0.83 (0.48, 1.45)	0.85 (0.49, 1.48)	0.88 (0.51, 1.53)
Random effect			
Level 2 variance-Individual	0.94 (0.64, 1.37)	0.95 (0.65, 1.38)	0.88 (0.60, 1.28)
Level 3 variance-Community	1.03 (0.69, 1.54)	1.04 (0.69, 1.55)	1.08 (0.72, 1.61)

Abbreviation: S, sixtile; CDGI-E, China Elderly Dietary Guidelines Index; PA, physical activity; WC, waist circumference. ^a^
*p* for trend > 0.05. ^1^ The median values were assigned to sixtiles of the CDGI-E score and used to fit the model as a continuous term for testing linear trends. ^2^ We adjusted for age, educational level, geographic region, urbanicity index, household income, and energy intake (Model 1), PA and smoking status (Model 2), and baseline WC (Model 3).

**Table 5 nutrients-14-02623-t005:** OR (95% CI) of the risk of abdominal obesity across the CDGI-E scores.

	Men			Women		
	Model 1 ^2^	Model 2 ^2^	Model 3 ^2^	Model 1 ^2^	Model 2 ^2^	Model 3 ^2^
Fixed effect						
CDGI-E scores ^1^						
S1	1.00 (1.00, 1.00) ^a^	1.00 (1.00, 1.00) ^a^	1.00 (1.00, 1.00) ^a^	1.00 (1.00, 1.00) ^a^	1.00 (1.00, 1.00) ^a^	1.00 (1.00, 1.00) ^a^
S2	1.01 (0.63, 1.61)	0.99 (0.62, 1.58)	1.08 (0.68, 1.72)	0.96 (0.67, 1.39)	0.96 (0.67, 1.39)	0.99 (0.68, 1.42)
S3	1.24 (0.79, 1.95)	1.21 (0.77, 1.90)	1.25 (0.80, 1.96)	1.05 (0.73, 1.52)	1.04 (0.72, 1.50)	1.08 (0.75, 1.56)
S4	0.96 (0.61, 1.52)	0.94 (0.59, 1.49)	0.95 (0.60, 1.50)	1.15 (0.79, 1.67)	1.13 (0.78, 1.64)	1.16 (0.80, 1.68)
S5	0.85 (0.53, 1.36)	0.82 (0.51, 1.32)	0.88 (0.55, 1.40)	1.16 (0.79, 1.70)	1.14 (0.78, 1.66)	1.29 (0.88, 1.89)
S6	0.90 (0.56, 1.45)	0.87 (0.54, 1.41)	0.93 (0.58, 1.49)	0.61 (0.41, 0.91) *	0.60 (0.40, 0.89) *	0.62 (0.41, 0.92) *
Random effect						
Level 2 variance-Individual	3.16 (2.16, 4.16) ***	2.99 (2.02, 3.96) ***	2.27 (1.45, 3.08) ***	2.33 (1.50, 2.96) ***	2.14 (1.44, 2.85) ***	1.59 (1.00, 2.18) ***
Level 3 variance-Community	0.68 (0.19, 1.17) **	0.65 (0.17, 1.12) **	0.47 (0.08, 0.87) **	0.65 (0.04, 0.60) *	0.034 (0.06, 0.61) *	0.27 (0.04, 0.52) *

Abbreviation: S, sixtile; CDGI-E, China Elderly Dietary Guideline Index; PA, physical activity; WC, waist circumference. ^a^
*p* for trend > 0.05, * *p* < 0.05, ** *p* < 0.01, *** *p* < 0.001. ^1^ The median values were assigned to sixtiles of the CDGI-E score and used to fit the model as a continuous term for testing linear trends. ^2^ We adjusted for age, educational level, geographic region, urbanicity index, household income, and energy intake (Model 1), PA and smoking status (Model 2), and baseline WC (Model 3).

## Data Availability

Data sharing is not applicable to this article.

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
