# Peer review of "High Diet Quality Is Linked to Low Risk of Abdominal Obesity among the Elderly Women in China"

_nutrients, 2022, doi:10.3390/nu14132623_

Round 1
Reviewer 1 Report
The authors present an interesting research topic that examines the association between diet quality and obesity among older adults using data from China Health and Nutrition Survey. In particular, the authors study a link between China Diety Guideline Index-Elderly with abdominal obesity. They did find an association between diet quality and abdominal obesity in Chinese older adults women, but not in men. This manuscript gives insight into the importance of consuming improved diet quality for better health outcomes. It is a well-written manuscript and has substantial merit. Whereas the topic itself is important, some issues warrant revision. Please see below:
· Abstract: no need to mention “Background”, “Methods”, “Results” and “Conclusion”. Policy implications or potential mechanisms of an association between CDGI-E and reduced obesity in women could be added at the end of the abstract
· Introduction: The introduction does not provide sufficient background information about this study. Authors can make a nice story in the introduction to motivate the research topic. Since what people eat is directly associated with their diet-related health and nutritional outcomes, particularly among low-income households. Authors could argue that a better understanding of the link between diet quality and health outcomes is so important that it helps policymakers make an informed decision to curb the problem of diet-induced health-related issues. Please refer to the following manuscripts and come up with a nice background. Authors could take these references to highlight the relationship in the discussion.
1. Dhakal, C.K.; Khadka, S. Heterogeneities in Consumer Diet Quality and Health Outcomes of Consumers by Store Choice and Income. Nutrients 2021, 13, 1046. https://doi.org/10.3390/nu13041046
2. GBD 2015 Obesity Collaborators. (2017). Health effects of overweight and obesity in 195 countries over 25 years. New England Journal of Medicine, 377(1), 13-27.
· The manuscript needs to highlight the innovation of the research and clarify the shortcomings of the existing research (gap in the literature) and the contribution of this research.
· The conclusion part of the manuscript is mostly a description of the research results. The author should add some detailed discussions on policy formulation purposes and the practical significance of the research to make the research more meaningful. The conclusion should be a separate healding.
Author Response
Response to Reviewer 1 Comments
Point 1: Abstract: no need to mention “Background”, “Methods”, “Results” and “Conclusion”. Policy implications or potential mechanisms of an association between CDGI-E and reduced obesity in women could be added at the end of the abstract.
Response 1: Thank you for your comment. “Background”, “Methods”, “Results” and “Conclusion” has been deleted from the abstract as you suggested, and "Our findings will help to improve the understanding of the relationship between the overall effect of diet and health. It may provide a new avenue for obesity intervention policy formulation from the aspect of improving the overall dietary quality" has been added to the last sentence.
Point 2: Introduction: The introduction does not provide sufficient background information about this study. Authors can make a nice story in the introduction to motivate the research topic. Since what people eat is directly associated with their diet-related health and nutritional outcomes, particularly among low-income households. Authors could argue that a better understanding of the link between diet quality and health outcomes is so important that it helps policymakers make an informed decision to curb the problem of diet-induced health-related issues. Please refer to the following manuscripts and come up with a nice background. Authors could take these references to highlight the relationship in the discussion.
- Dhakal, C.K.; Khadka, S. Heterogeneities in Consumer Diet Quality and Health Outcomes of Consumers by Store Choice and Income. Nutrients2021, 13, 1046. https://doi.org/10.3390/nu13041046
- GBD 2015 Obesity Collaborators. (2017). Health effects of overweight and obesity in 195 countries over 25 years. New England Journal of Medicine, 377(1), 13-27.
Response 2: Thank you so much for your detailed review. After carefully reading the articles you recommended, we have added the following contents in the introduction to increase the readability and scientificity of the article:
“At present, the food environment is changing rapidly. People eat out and eat processed food more frequently. Nutritional problems such as excessive intake of macronutrients, lack of micronutrients, and food insecurity persist[9]. There are some problems in the elderly, such as excessive intake of energy, saturated fat, and meat, and insufficient intake of protein and micronutrients[10]. An American study shows that from 2001 to 2018, the dietary quality of the elderly gradually decreased[11]. The nutritional imbalance of the elderly in China is also prominent, which is mainly manifested in excessive intake of meat and edible oil, but insufficient intake of vegetables, fruits, soybeans, and dairy products[12]. Some studies show that dietary energy density, the consumption of processed food, red meat intake, and other dietary factors were associated with the prevalence of obesity, cardiovascular diseases, type 2 diabetes, etc.[10-12]. ”
In addition, the article you recommended has been cited in the article. Thank you again for your guidance.
Point 3: The manuscript needs to highlight the innovation of the research and clarify the shortcomings of the existing research (gap in the literature) and the contribution of this research.
Response 3: Thank you for your comment. We revise the advantages section as follows:
“Compared with the research on other age groups, the overweight and obesity of the elderly do not pay enough attention to the diet quality and health-related issues, which hinders the formulation of nutrition policies for the obesity the elderly. A systematic review showed that only one of 479 articles was about the elderly, and very few studies have been followed up for more than ten years[47]. The strengths of our research include the prospective design, large sample sizes of a special population, precise dietary data collection using consecutive 3-d 24-h recalls combined with household weighing for cooking oil and condiments, the use of CDGI-E to measure overall diet quality and its association with abdominal obesity, the use of multilevel mixed effect model to handle the hierarchy and unbalance data structure allowing non-independence of repeated measurement, and adjustment for comprehensive latent covariables and potential heterogeneity. This study was followed up for more than 20 years to reflect the long-term effect of dietary patterns on body weight. CDGI-E was used based on the critical diet-related recommendations of the CDG-2016 to access overall diet quality, which is more suitable for Chinese people. ”
Point 4: The conclusion part of the manuscript is mostly a description of the research results. The author should add some detailed discussions on policy formulation purposes and the practical significance of the research to make the research more meaningful. The conclusion should be a separate healding.
Response 4: Thank you for your comment. We revise the conclusion as follows:
”The diet quality of Chinese elderly women, measured by the CDGI-E, was independently associated with the risk of abdominal obesity. Therefore, attention should be paid to the dietary quality of the elderly, especially the elderly women. It is necessary to strengthen the research on the dietary factors related to overweight and obesity in the elderly, to provide the scientific basis for dietary intervention. It is suggested that the elderly should be fully considered in the formulation of obesity prevention and control policies. We should do a good job in science popularization, education, and nutrition intervention aimed at improving the dietary quality of the elderly. Further study to explore the CDGI-E contributing to other cardio-metabolic risk factors for verifying its practicability. ”

Reviewer 2 Report
This manuscript consists of useful contents that can analyze the health status of the elderly in China and develop indicators for improving their health.
1. First, there must be enough explanation about “sixtiles” on a scientific basis.
2. The references may be updated.
Author Response
Response to Reviewer 2 Comments
Point 1: First, there must be enough explanation about “sixtiles” on a scientific basis.
Response 1: Thank you for your comment. Because the dietary quality of most people in the study was low, to better compare the effects of different dietary quality on obesity, we divided the participants into sixtiles by CDGI-E score and gender instead of the traditional quartile. The above-related contents have been added in the “Statistical analysis” section.
Point 2: The references may be updated.
Response 2: Thank you so much for your detailed review. Our updated references are as follows:
- GBD 2015 Obesity Collaborators. Health effects of overweight and obesity in 195 countries over 25 years. New England Journal of Medicine 2017, 377(1), 13-27.
- Wells, J.C.K.; Marphatia, A. A.; Amable, G.; Siervo, M.; Friis, H.; Miranda, J. J.; Haisma, H.H.; Raubenheimer, D. The future of human malnutrition: rebalancing agency for better nutritional health. Global Health2021, Oct 9;17(1):119. doi: 10.1186/s12992-021-00767-4.
- Roberts, S. B.; Silver, R. E.; Das, S.K.; Fielding, R.A.; Gilhooly, C.H.; Jacques, P.F.; Kelly, J.M.; Mason, J.B.; McKeown, N.M.; Reardon, M.A.; Rowan, S.; Saltzman, E.; Shukitt-Hale, B.; Smith, C.E.; Taylor, A.A.; Wu, D.; Zhang, F.F.; Panetta, K.; Booth, S. Healthy Aging-Nutrition Matters: Start Early and Screen Often. Adv Nutr2021, Jul 30;12(4):1438-1448. doi: 10.1093/advances/nmab032.
- Long, T.; Zhang, K.; Chen, Y.; Wu, C. Trends in Diet Quality Among Older US Adults From 2001 to 2018. JAMA Netw Open2022, Mar 1;5(3):e221880. doi: 10.1001/jamanetworkopen.2022.1880.
- Wang, L.; Ouyang, Y.; Jiang, H.; Zhang, B.; Wang, H.; Zhang, J.; Du, W.; Niu, R.; Wang, Z. Secular trends in food intakes among the elderly aged 60 and older in nine provinces in China from 1991 to 2015. Wei Sheng Yan Jiu2022, Jan;51(1):24-31. Chinese. doi: 10.19813/j.cnki.weishengyanjiu.2022.01.005.
- Takeda, Y.; Fujihara, K.; Nedachi, R.; Ikeda, I.; Morikawa, S. Y.; Hatta, M.; Horikawa, C.; Kato, M.; Kato, N.; Yokoyama, H; Kurihara, Y.; Miyazawa, K.; Maegawa, H.; Sone, H. Comparing Associations of Dietary Energy Density and Energy Intake, Macronutrients with Obesity in Patients with Type 2 Diabetes (JDDM 63). Nutrients 2021, Sep 11;13(9):3167. doi: 10.3390/nu13093167.
- Dhakal, C.K.; Khadka, S. Heterogeneities in Consumer Diet Quality and Health Outcomes of Consumers by Store Choice and Income. Nutrients2021, 13, 1046. https://doi.org/10.3390/nu13041046
- Daneshzad, E.; Askari, M.; Moradi, M.; Ghorabi, S.; Rouzitalab, T.; Heshmati, J.; Azadbakht, L. Red meat, overweight and obesity: A systematic review and meta-analysis of observational studies. Clinical Nutrition ESPEN 2021. doi:10.1016/j.clnesp.2021.07.028
- 9. Riseberg,E.; Tamez, M.; Tucker, K. L.; Rodriguez; Orengo, J. F.; Mattei, Associations between diet quality scores and central obesity among adults in Puerto Rico. J Hum Nutr Diet 2021, Dec;34(6):1014-1021. doi: 10.1111/jhn.12873..
- 10. Cacau, L. T.; Benseñor, I. M.; Goulart, A. C.; Cardoso, L. O.; Lotufo, P. A.; Moreno, L.A.; Marchioni, D.M. Adherence to the Planetary Health Diet Index and Obesity Indicators in the Brazilian Longitudinal Study of Adult Health (ELSA-Brasil). Nutrients 2021, Oct 20;13(11):3691. doi: 10.3390/nu13113691.
- 11. Kadam, I.; Neupane, S.; Wei, J.; Fullington, L. A.; Li, T.; An, R.; Zhao, L.; Ellithorpe, A.; Jiang, X.; Wang, L.; A Systematic Review of Diet Quality Index and Obesity among Chinese Adults. Nutrients 2021, Oct 11;13(10):3555. doi: 10.3390/nu13103555.

Reviewer 3 Report
In the presented work, two goals can be distinguished, one is to define an index of the quality of the diet for the population of Chinese elderly (CDGI-E). On the other hand, the second aim is to present the relationship of this indicator with the occurrence of excess body weight and central obesity in study group.
Information on the dietary quality indicators used in national surveys should be moved from introduction to the discussion. However, the introduction lacks information on the presentation of the quality of nutrition of elderly people in China in relation to epidemiological data.
Materials and Methods
Please explain from which year the data on the waist circumference was collected (2 different dates are given) in line 90-92.
Construction of the CDGI-E. Please clarify what is included in the term edible oil to be limited ? Is it about animal fats? Whether it is the fats used in food preparation ?
Results
There is no information on the maximum age and average age of the study group of elderly people.
There is also a lack of data on the interpretation of BMI and the interpretation of WC according to the adopted criteria.
In the case of older people, BMI values will be interpreted differently than for adult population (for people over 65 years of age the BMI is normal within the values 23.5 - 29.9). No reference in discussion to the BMI assessment criterion in study group.
Does the Total score in Table 3 apply to CDGI-E results?
The discussion needs to be supplemented and structured in relation to the assumed goals of the work. The authors in the discussion do not refer to other studies conducted in the elderly population in relation to the quality of diet and the prevalence of central obesity.
Author Response
Response to Reviewer 1 Comments
Point 1: Information on the dietary quality indicators used in national surveys should be moved from introduction to the discussion. However, the introduction lacks information on the presentation of the quality of nutrition of elderly people in China in relation to epidemiological data.
Response 1: Thank you for your comment. The related contents of dietary quality indicators have been put into the discussion according to your suggestion. The introduction added the epidemiological characteristics of nutrition-related problems of the elderly in China and other countries, the relationship between nutrition and diseases, and the dietary environment.The specific contents are as follows:
“At present, the food environment is changing rapidly. People eat out and eat processed food more frequently. Nutritional problems such as excessive intake of macronutrients, lack of micronutrients, and food insecurity persist[9]. There are some problems in the elderly, such as excessive intake of energy, saturated fat, and meat, and insufficient intake of protein and micronutrients[10]. An American study shows that from 2001 to 2018, the dietary quality of the elderly gradually decreased[11]. The nutritional imbalance of the elderly in China is also prominent, which is mainly manifested in excessive intake of meat and edible oil, but insufficient intake of vegetables, fruits, soybeans, and dairy products[12]. Some studies show that dietary energy density, the consumption of processed food, red meat intake, and other dietary factors were associated with the prevalence of obesity, cardiovascular diseases, type 2 diabetes, etc.[13-15]. ”
Point 2:Please explain from which year the data on the waist circumference was collected (2 different dates are given) in line 90-92.
Response 2: Thank you so much for your detailed review. We made a mistake. It should be counted as 1993. The related content has been revised in the article.
Point 3: Construction of the CDGI-E. Please clarify what is included in the term edible oil to be limited ? Is it about animal fats? Whether it is the fats used in food preparation ?
Response 3: Thank you for your comment. Edible oil includes vegetable oil and animal fat, which are used in cooking. The above content has been added in the “Dietary quality measurement ” section.
Point 4: There is no information on the maximum age and average age of the study group of elderly people.
Response 4: Thank you for your comment. The average age of the subjects was (64.88±5.42) years old, ranging from 60.00 to 93.83 years old. The relevant content is added in the first paragraph of the result section.
Point 5: There is also a lack of data on the interpretation of BMI and the interpretation of WC according to the adopted criteria.
Response 5: Thanks. We supplemented the endpoint overweight/ general obesity and abdominal obesity rates of the subjects. Table S1 shows the endpoint overweight/ general obesity and abdominal obesity rates of participants by gender and CDGI-E. Because the baseline excludes overweight/ general obesity and abdominal obesity people, the baseline results only supplement BMI and waist circumference. We added the following to the first paragraph of the result:
”At the baseline, the subjects' BMI was (21.07±2.62)kg/cm2 and waist circumference was (75.25±7.04)cm. At the endpoint, the abdominal obesity rate was 30.68%, and the overweight/ general obesity rate was 13.03%(Table S1). ”
Point 6: In the case of older people, BMI values will be interpreted differently than for adult population (for people over 65 years of age the BMI is normal within the values 23.5 - 29.9). No reference in discussion to the BMI assessment criterion in study group.
Response 6: Thank you for your comment. We think it is very meaningful to explain your questions in the discussion section. In this study, 68.75% of the elderly are under 65 years old, and only 0.94% of the elderly have BMI>29.9. Studies have shown that BMI≥25 in the elderly will also increase the risk of type 2 diabetes, cardiovascular diseases, and cancer[10]. In order to better compare the influence of dietary quality on BMI, we have chosen the criteria of WHO Guidelines for adults, and BMI≥25.0 is defined as overweight/ general obesity. The above content has been added in the fifth paragraph of the discussion.
Point 7: Does the Total score in Table 3 apply to CDGI-E results?
Response 7: Thank you for your comment. Total score in Table 3 is CDGI-E results. Through your suggestion, we found that there are some ambiguities in our expression., so we changed “Total score” to “CDGI-E scores”. We have also checked other parts of the article, which have been revised according to your suggestion.
Point 8: The discussion needs to be supplemented and structured in relation to the assumed goals of the work. The authors in the discussion do not refer to other studies conducted in the elderly population in relation to the quality of diet and the prevalence of central obesity.
Response 8: Thanks. Because there are few studies on the diet quality and central obesity of the elderly, only other studies on the diet quality and central obesity of adults are compared in the discussion. Your suggestion is very important. Therefore, we have supplemented a document and added an introduction that is less researched at present. Specific content is as follows:
1.There are few studies on the dietary quality of the elderly and abdominal obesity. In line with the results from other countries on adults, our study also observed the inverse association of the CDGI-E scores with the risk of abdominal obesity among the older women after adjusting for all potential confounders[13,16].
2.Compared with the research on other age groups, the overweight and obesity of the elderly do not pay enough attention to the diet quality and health-related issues, which hinders the formulation of nutrition policies for the obesity the elderly. A systematic review showed that only one of 479 articles was about the elderly, and very few studies have been followed up for more than ten years[47].
